# Predicting Mortality in Atrial Fibrillation Patients Treated with Direct Oral Anticoagulants: A Machine Learning Study Based on the MIMIC-IV Database

**DOI:** 10.3390/jcm14113697

**Published:** 2025-05-25

**Authors:** Łukasz Ledziński, Elżbieta Grześk, Małgorzata Ledzińska, Grzegorz Grześk

**Affiliations:** 1Department of Cardiology and Clinical Pharmacology, Collegium Medicum in Bydgoszcz, Nicolaus Copernicus University in Toruń, 85-168 Bydgoszcz, Poland; 503345@doktorant.umk.pl (Ł.L.); g.grzesk@cm.umk.pl (G.G.); 2Department of Pediatrics, Hematology and Oncology, Collegium Medicum in Bydgoszcz, Nicolaus Copernicus University in Toruń, 85-094 Bydgoszcz, Poland; 3Department of Rheumatology and Connective Tissue Diseases, Collegium Medicum in Bydgoszcz, Nicolaus Copernicus University in Toruń, 85-168 Bydgoszcz, Poland; malgorzata.ledzinska.93@gmail.com

**Keywords:** machine learning, atrial fibrillation, direct oral anticoagulants, cardiology

## Abstract

**Background/Objectives:** Atrial fibrillation (AF) is a common arrhythmia linked to increased mortality and significant healthcare burden, especially in the elderly. Direct Oral Anticoagulants (DOACs) are crucial for stroke prevention in AF, offering benefits over traditional vitamin K antagonists. Despite scoring systems like HATCH and CHA2DS2-VASc, their predictive ability for mortality in AF patients is limited. This study aims to use machine learning to predict mortality within six months of hospital discharge in AF patients treated with DOACs. **Methods:** Using the MIMIC-IV database, data from 6431 AF patients were analyzed. Feature selection was done with the LASSO algorithm. Five machine learning models were built: Logistic Regression, Random Forest, XGBoost, LightGBM, and AdaBoost, using 27 features. The top two models were tested on a separate dataset. SHAP values explained model predictions and feature importance. **Results:** The best model, LightGBM, achieved an AUC of 0.886, accuracy of 0.862, sensitivity of 0.913, and specificity of 0.859. SHAP values highlighted the importance of length of hospital stay, ICU duration, and comorbidities. The model’s interpretability allows for identifying individual patient risk factors, applicable in clinical practice. **Conclusions:** This study demonstrates that machine learning models effectively predict mortality in AF patients treated with DOACs, potentially enhancing personalized patient care.

## 1. Introduction

Atrial fibrillation (AF) is a common supraventricular arrhythmia associated with an increased risk of mortality. It is characterized by a rapid and chaotic heart rhythm, resulting in disrupted blood flow and potentially leading to serious complications such as stroke and heart failure, which adversely affect patient longevity and quality of life [1,2,3]. The high prevalence of AF imposes a significant burden on the healthcare system [4,5]. Additionally, the aging population significantly increases the incidence of this condition [5,6]. Direct Oral Anticoagulants (DOACs) have become a cornerstone in the management of atrial fibrillation (AF), particularly for stroke prevention [7]. Unlike traditional vitamin K antagonists (VKAs) such as warfarin, DOACs offer several advantages, including predictable pharmacokinetics, fewer dietary restrictions, and a lower risk of intracranial hemorrhage [8]. DOACs, which include agents such as dabigatran, rivaroxaban, apixaban, and edoxaban, directly inhibit specific clotting factors (thrombin or factor Xa), providing effective anticoagulation with a more favorable safety profile. Clinical trials and real-world studies have demonstrated that DOACs are at least as effective as VKAs in preventing stroke and systemic embolism in patients with non-valvular AF, with a reduced risk of major bleeding [9]. Consequently, current guidelines recommend DOACs as the preferred anticoagulant therapy for most patients with AF, barring specific contraindications. Artificial intelligence has found widespread application in various fields, including medicine, and particularly in cardiology, where its capabilities are widely used [10,11]. Currently used scoring systems such as HATCH, HAVOC, and CHA2DS2-VASc have shown limited ability to predict mortality in patients with atrial fibrillation (AF), as demonstrated in several studies [12,13,14]. Attempts to use machine learning for predicting mortality in patients with atrial fibrillation (AF) have been undertaken by various researchers [15,16,17]. Given the increasing number of patients with AF and the widespread use of DOACs in this patient group, we decided to utilize machine learning tools to predict mortality within six months of hospital discharge in this population.

## 2. Materials and Methods

The study focuses on utilizing retrospective data from the Medical Information Mart for Intensive Care (MIMIC)-IV version 3.1 database [18,19]. This is a large, anonymized database from the Beth Israel Deaconess Medical Center in Boston, MA, containing data for 364,627 patients from 2008 to 2022. Access to the MIMIC database was obtained through a multi-step process to ensure ethical and responsible use of the data. First, researchers completed the Collaborative Institutional Training Initiative (CITI) “Data or Specimens Only Research” course, which provides essential training on the ethical considerations and responsibilities associated with handling sensitive health data. Following this, a Data Use Agreement (DUA) was signed and submitted, outlining the terms and conditions for using the MIMIC data in compliance with all applicable regulations. Finally, credentialing was obtained through PhysioNet, the platform hosting the MIMIC database [20]. Upon successful completion of these steps, access to the MIMIC database was granted for research purposes. Patients with atrial fibrillation (AF) (n = 32,026) were selected for the study, identified by ICD codes [‘42731’, ‘I4891’, ‘I48’, ‘I480’, ‘I481’, ‘I4811’, ‘I4819’, ‘I482’, ‘I4820’, ‘I4821’, ‘I489’], who initiated treatment with DOACs (n = 6431) (dabigatran/rivaroxaban/apixaban/edoxaban) during hospitalization. Patients who died during their hospital stay were excluded from the cohort. Information was selected on medical history, hospital stay, Charlson Comorbidity Index (CCI) score, demographic details such as age, gender, race, marital status, hospital and ICU stay, administered medications, microbiology test results, performed procedures, diagnostic codes, and laboratory results, resulting in a total of 600 variables.

Data cleaning was conducted by removing all variables with more than 10% missing values. Feature engineering was performed to extract information on the total ICU length of stay, the total number of diagnoses, hospital length of stay, the reason for hospital admission (whether it was related to a procedure or observation), the absence of performed procedures, negative microbiology test results, heparin treatment, and readmission within 60 days of discharge. Patient age was initially divided into 10-year intervals and then encoded in ascending order. Features such as ‘Gender’, ‘Congestive Heart Failure’, ‘Dementia’, ‘Diabetes without Complications’, ‘Paraplegia’, ‘Metastatic Solid Tumor’, ‘Readmission within 60 days’, ‘Marital Status: Widowed’, ‘Observation Admission’, ‘Surgical Admission’, ‘Microbiology: Negative’, ‘No applied procedures’, ‘Heparin’, and ‘Obesity’ were encoded as binary variables (0/1).

The remaining features were continuous variables: ‘Anion Gap’, ‘Bicarbonate’, ‘Hemoglobin’, ‘Potassium’, ‘Sodium’, ‘Urea Nitrogen’, ‘White Blood Cells’, ’Creatinine’, ‘Sum of ICU Length of Stay’, ‘Number of Diagnoses’, ‘Charlson Comorbidity Index’, and ‘Hospital Days’. All continuous variables were standardized prior to the application of Logistic Regression (LR) and the Least Absolute Shrinkage and Selection Operator (LASSO) algorithm. Patient records containing missing values were excluded from the dataset. Missing data are a common challenge in clinical datasets and can impact the validity and generalizability of predictive models. Several imputation strategies are available, including simple approaches such as mean or median substitution, as well as more sophisticated methods like k-nearest neighbors (KNN) imputation and MICE. Simple imputation methods are computationally efficient but may underestimate variability and distort associations between variables. KNN imputation leverages the similarity between observations, potentially preserving local data structure, but can be sensitive to outliers and the choice of k. MICE, an advanced iterative technique, models each variable with missing values as a function of other variables, generating multiple plausible datasets and thus accounting for imputation uncertainty. In this study, we explored multiple imputation using MICE in addition to listwise deletion. However, the application of MICE did not result in a meaningful improvement in model performance or alter the main findings. Given the lack of demonstrable benefit from imputation, we opted for listwise deletion to maximize methodological transparency and interpretability while minimizing the risk of introducing imputation-related bias. This approach is further supported by the absence of evidence for systematic differences between complete and incomplete cases. Only data from the initial hospitalization when atrial fibrillation (AF) was first diagnosed were considered. The class distribution was 6086:345. The prepared dataset was then split into training and testing sets in an 80:20 ratio. Feature selection was performed using the LASSO algorithm, resulting in the selection of 27 features for model construction. LASSO is a regularization technique that enhances both the prediction accuracy and interpretability of regression models by imposing a constraint on the absolute values of the model parameters, effectively shrinking some coefficients to zero. This method is particularly advantageous in high-dimensional datasets, where it aids in mitigating overfitting and improving model generalization by selecting a subset of the most informative features. Consequently, LASSO is widely employed for feature selection, as it systematically identifies and retains only the most relevant predictors, thereby simplifying the model and enhancing its performance. A machine learning model was developed using the eXtreme Gradient Boosting (XGBoost), Adaptive Boosting (AdaBoost), Light Gradient-Boosting Machine (LightGBM), Random Forest (RF), and LR algorithms to predict patient mortality from any cause within six months of hospital discharge. XGBoost is an advanced gradient- boosting algorithm that enhances performance and speed through optimized implementation, making it highly effective for structured data analysis. AdaBoost is an ensemble learning technique that improves classification accuracy by iteratively adjusting the weights of weak classifiers, thereby focusing on previously misclassified instances. LightGBM is a gradient-boosting framework that leverages tree-based learning algorithms, optimized for high efficiency and scalability, particularly suitable for large-scale data processing. Random Forest is an ensemble method that constructs multiple decision trees during training and aggregates their predictions, known for its robustness and ability to reduce overfitting. Logistic Regression is a statistical method that models the probability of a binary outcome using a logistic function, valued for its simplicity and interpretability in binary classification tasks.

During training, internal validation was performed using RepeatedStratifiedKFold (n_splits = 5, n_repeats = 5) to ensure the accuracy and robustness of the model. Hyperparameter tuning for all models was conducted using the HyperOpt library [21]. HyperOpt is a powerful optimization library in Python designed for hyperparameter tuning, which uses Bayesian optimization and other search algorithms to efficiently explore the hyperparameter space. It is particularly useful for improving model performance by finding the optimal set of hyperparameters in a computationally efficient manner. The top two models with the best internal validation performance were used to make predictions on the test set.

To assess the model’s quality, standard metrics for binary classification were utilized, such as accuracy, specificity, sensitivity, the Matthews correlation coefficient, and area under the ROC curve.

Accuracy denotes the proportion of correctly classified instances among all instances. It reflects the overall performance of the model and is straightforward to interpret. However, in the context of imbalanced datasets, accuracy can be deceptive as it may not accurately represent the model’s performance on the minority class.Accuracy = (TP + TN)/(TP + TN + FP + FN).(1)

Specificity, or the true negative rate, measures the model’s ability to correctly identify negative instances. It is crucial in situations where minimizing Type I errors (false positives) is essential. High specificity ensures that the majority of actual negative cases are accurately identified by the model.Specificity = TN/(TN + FP).(2)

Sensitivity, also known as recall, assesses the model’s ability to correctly identify positive instances. It is vital in scenarios where minimizing Type II errors (false negatives) is important. High sensitivity ensures that most actual positive cases are detected by the model.Sensitivity = TP/(TP + FN).(3)

The Matthews Correlation Coefficient (MCC) is a balanced measure of binary classification performance that considers all four values of the confusion matrix. It is particularly useful for imbalanced datasets. The MCC ranges from −1 to +1, where +1 indicates perfect prediction, 0 indicates random prediction, and −1 indicates total disagreement between prediction and observation.MCC = ((TP·TN) − (FP·FN))/√((TP + FP)·(TP + FN)·(TN + FP)·(TN + FN)).(4)

The Receiver Operating Characteristic—Area Under the Curve (ROC–AUC) represents the area under the curve that illustrates the relationship between sensitivity and 1-specificity across various decision thresholds. It evaluates the model’s discriminative ability; a higher value indicates a better capability to distinguish between classes:True Positive (TP): Correctly predicted death of patient;True Negative (TN): Correctly predicted survival of patient;False Positive (FP): Incorrectly predicted survival of patient, type I error;False Negative (FN): Incorrectly predicted death of patient, type II error.

All computations presented in this study were conducted in Python 3.10.12 on a CPU featuring 24 threads at 4.5 GHz and 64 GB of RAM. Packages versions: pandas (2.1.4), numpy (1.26.4), hyperopt (0.2.7), shap (0.46.0), scikit-learn (1.4.2), xgboost (2.1.3), lightgbm (4.5.0).

## 3. Results

### 3.1. The General Characterization of Patients Group

The study population consisted of patients with atrial fibrillation (AF) who were treated with Direct Oral Anticoagulants (DOACs) identified from the MIMIC-IV database, with baseline characteristics compared between survivors and non-survivors. The baseline characteristics of patients were compared between the group of survivors and non-survivors. Categorical variables were expressed as count and frequency, compared using the chi-square test. Continuous variables were presented as mean ± standard deviation (SD) or median and first and third quartiles (Q1, Q3). Continuous variables were analyzed using Mann–Whitney U test. A two-side test result with *p*-value < 0.05 was considered statistically significant. Table 1, below, summarizes the statistics of continuous and categorical variables used to build the model.

### 3.2. Machine Learning

ROC–AUC curves were presented for all five algorithms used during training (see Figure 1). As initially observed, the results obtained by LightGBM, XGBoost, and Random Forest are very similar on the ROC curves, suggesting comparable effectiveness of these algorithms in the analyzed task. The primary factors determining the final algorithm selection were MCC and AUC, as these metrics effectively evaluate the quality of the classification model in the context of a highly imbalanced dataset. Table 2 summarizes all metrics for each model, along with the 95% confidence intervals (CI) for the results obtained during internal cross-validation.

Table 3 presents the prediction results on the test set obtained by the two best-performing models during training. The similar results achieved by the closely related algorithms, LightGBM and XGBoost, are not surprising; both algorithms are based on gradient boosting and share many underlying principles and techniques [22,23]. However, the results obtained by LightGBM were superior. All metrics showed higher values, with particular emphasis on the model’s high sensitivity. Given the importance of predicting patient mortality, maintaining high sensitivity is crucial, while also achieving satisfactory specificity results. The Z-test performed on the results obtained by the LightGBM and XGBoost models on the test data did not reveal statistically significant differences in sensitivity or specificity (*p* >> 0.05). Compared to the CHA_2_DS_2_-VASc score (mean 3.01, SD 1.318), which achieved an AUC of 0.639 (95% CI: 0.611–0.665) for mortality prediction, the results obtained by the presented models were substantially superior [24,25].

### 3.3. Shapley Values

To explain the obtained results, the SHAP library was used, which employs Shapley values to identify the features with the greatest impact on predictions both globally and locally [26,27]. A beeswarm plot, as presented in Figure 2, visualizes the impact of each feature on the model’s predictions using SHAP values. Each point on the plot represents a SHAP value for a feature and an individual prediction. The position on the x-axis indicates the magnitude and direction of the feature’s impact on the prediction, while the color represents the feature value. This plot provides a comprehensive overview of which features are most influential in the model’s decision-making process and how they contribute to the predictions. As shown, the most important features include the length of stay in the ICU and the hospital, as well as comorbidities. Besides laboratory test results, these factors directly inform us about the clinical condition of patients with AF. Additional features that improved the model’s performance include the type of admission, the absence of performed procedures, and negative microbiology test results. Age and gender also have a significant impact on the model’s predictions. The only medication among the selected 27 features is heparin treatment, which inhibits the blood coagulation process primarily by activating antithrombin (AT), a potent inhibitor of thrombin and a weaker inhibitor of other plasma coagulation factors (Xa, IXa, XIa, XIIa).

## 4. Discussion

There is a lack of machine learning models capable of predicting mortality in patients using DOACs for AF. According to ESC recommendations, these drugs are the first choice for preventing thromboembolic events in this patient group [28]. The use of DOACs in the treatment of patients with AF is not only associated with significant therapeutic benefits but has also been shown to be safe in numerous studies, particularly with respect to the risk of major bleeding and intracranial hemorrhage. For instance, the XANTUS study—a prospective, real-world registry including over 6700 patients with AF treated with rivaroxaban—reported a major bleeding rate of 2.1 per 100 patient-years and confirmed a low incidence of fatal bleeding events [29]. Similarly, recent data from an Italian multicenter registry demonstrated that DOACs, including rivaroxaban, were safe and effective in routine clinical practice, with no major bleeding complications observed during a 12-month follow-up period [30]. These findings are further corroborated by large-scale analyses from Danish cohorts, which have shown that DOACs are associated with a lower risk of death and major bleeding compared to warfarin, supporting their favorable safety profile in real-world settings [31]. Since the MIMIC-IV database also contains records from before 2010, a period before the widespread use of DOACs in AF patients, the number of patients in this group is relatively low. A significant challenge is the high imbalance of the dataset, which necessitates the use of algorithms capable of handling such data and the selection of specific quality metrics. Mortality studies based on this database often focus on patients in the ICU, primarily due to the abundance of data for these patients [32,33,34]. In the context of the MIMIC database, ICU is considered as a stay in any intensive care unit, without distinguishing whether it was an MICU (Medical Intensive Care Unit), an SICU (Surgical Intensive Care Unit), a CCU (Cardiac Care Unit), a CSRU (Cardiac Surgery Recovery Unit), or an NICU (Neonatal Intensive Care Unit). All these units are collectively referred to as ICU in the MIMIC database [19,35]. Developing such a model enables the implementation of appropriate clinical measures, such as closer monitoring of the patient’s condition to reduce the risk of death. The model is not limited to predicting death from a specific cause due to the limited scope of the data. An additional limitation is the availability of data from only one facility. Following the example of the MIMIC database, it would be beneficial for other large institutions to develop similar datasets. This would reduce data bias and improve the testing of developed models. Another important limitation is the use of a simplified preprocessing approach for handling missing data by removing incomplete records. While this reduces complexity and maintains a streamlined workflow, it introduces the risk of bias. Exploring the impact of various data imputation techniques would be an interesting direction for future research. Preliminary attempts to apply the Multiple Imputation by Chained Equations (MICE) method did not yield a significant improvement in results. Additionally, the study does not employ more complex algorithms or ensemble methods, which could potentially improve performance but often at the expense of model interpretability. The significant risk factors for mortality within six months identified by the model for this patient group can serve as a basis for further research, including clinical studies. In the future, this could lead to improved personalized treatment for AF patients. It is crucial to interpret the results correctly; the model indicating heparin does not imply that heparin increases the risk of death. Instead, it should be interpreted as a reflection of the clinical condition of the patient requiring such treatment according to current guidelines. It is important to remember the crucial role that proper adherence plays in the use of DOACs, which is influenced by various factors, including the patient’s age and social status. Effective communication between patients and healthcare providers is crucial for improving treatment adherence. This involves clearly explaining the disease and the necessity of the treatment, as well as understanding the patient’s lifestyle, physical and financial capabilities, and preferences. Involving the patient in the decision-making process regarding anticoagulant therapy leads to a better understanding and acceptance of the therapy goals [36]. The selected features serve as descriptors of the patient’s clinical status and should not be interpreted as independent markers equivalent to those obtained from clinical studies. Artificial intelligence models should be utilized by specialists who are capable of interpreting the presented results and using them as a supportive tool for assessing the clinical condition of a patient, merely highlighting important factors that may increase or decrease the risk of mortality in a given individual. Model indications, such as the identification of heparin use as significant, should not be regarded as recommendations for treatment modification. Instead, such findings should be interpreted as reflecting the clinical status of the patient, which may require specific therapeutic interventions based on current medical knowledge and established clinical guidelines. False-positive predictions may lead to unnecessary interventions, such as increased monitoring, additional medications, or invasive procedures. These actions could expose patients to avoidable side effects, increased healthcare costs, and psychological distress due to being labeled as high-risk. Overestimating mortality risk could divert limited healthcare resources (e.g., ICU beds or specialist attention) away from patients who genuinely need them, potentially compromising care for others. On the other hand, false-negative predictions may result in high-risk patients being overlooked for necessary interventions, leading to preventable adverse outcomes, including death. This undermines the primary goal of the predictive model. If patients or clinicians perceive the model as unreliable, it could erode trust in AI-based tools, hindering their adoption in clinical practice. Ethical considerations demand a careful balance between sensitivity (minimizing false negatives) and specificity (minimizing false positives). For mortality prediction, prioritizing sensitivity may be more ethical to ensure high-risk patients are not missed, but this must be weighed against the risks of overtreatment. Predictions should be used to support, not replace, clinical judgment. Patients must be informed about the limitations of the model, including the possibility of false predictions, to ensure they can make autonomous decisions about their care. False predictions may disproportionately affect certain patient groups if the model is biased due to imbalanced training data. This raises ethical concerns about fairness and equity in healthcare delivery. One of the features used by the model is obesity, which has been studied for its impact on the achieved concentrations of DOACs. This can result in either a lack of therapeutic effect or an increased risk of bleeding [37,38]. Another study focused on the direct impact of obesity on short- and medium-term mortality in critically ill patients with AF, demonstrating a lack of linear relationship between BMI and all-cause mortality [39].

In medical data analysis for predicting patient mortality, the application of various machine learning algorithms offers diverse benefits. Logistic Regression is a simple and interpretable model that allows for quick insights into feature significance and performs well with linearly separable data, serving as a solid baseline. AdaBoost, as a boosting method, effectively identifies difficult cases and improves model accuracy while reducing the risk of overfitting by limiting the depth of base classifiers. XGBoost, on the other hand, is known for its high performance and accuracy due to gradient optimization and built-in regularization, making it a popular choice for predictive tasks, especially with large datasets. LightGBM, an alternative to XGBoost, provides faster training and more efficient memory usage, which is crucial with a large number of variables and class imbalance. Random Forest offers stability and noise resistance by aggregating the results of multiple decision trees and allows for feature importance evaluation, supporting result interpretation in a clinical context. The selected set of algorithms is well established in numerous scientific studies utilizing binary classification to predict patient mortality across various conditions [40,41,42,43,44].

The use of Explainable AI (XAI) is essential for such solutions in medicine. Without the ability to trace the model’s reasoning, it is challenging to gain the trust of healthcare providers. SHAP values are effective in this context, both globally and locally. Local explanations for individual patients, indicating what drives the model’s predictions, not only enhance the trust of medical professionals but also enable personalized approaches. See Figure 3.

To the best of our knowledge, this is the first project aimed at predicting mortality in this specific patient group, and therefore we cannot compare our results with other studies. The MIMIC database itself is widely used for predicting patient mortality, including those with AF [45]. However, the target population in our study is different, making direct comparison of the obtained results impossible.

Although the model achieves highly satisfactory results, clinical judgment remains essential in interpreting and applying its predictions. It is important to remember that such models are tools to support daily work in clinical settings.

## 5. Conclusions

This study demonstrates the potential of using machine learning to predict mortality in AF patients using DOACs within six months of hospital discharge. The best results were achieved by the model based on the LightGBM algorithm. Utilizing AI-based tools can save clinical time and facilitate personalized disease management. However, further research is needed, preferably multi-center studies, to improve the model and validate it on entirely external datasets.

## Figures and Tables

**Figure 1 jcm-14-03697-f001:**
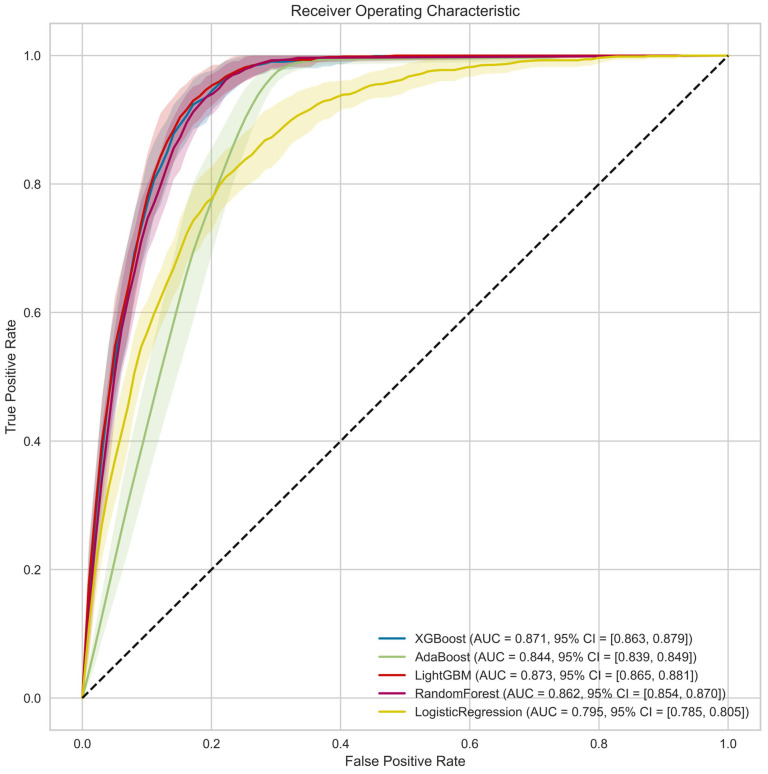
ROC curves visualizing performance of five different machine learning algorithms.

**Figure 2 jcm-14-03697-f002:**
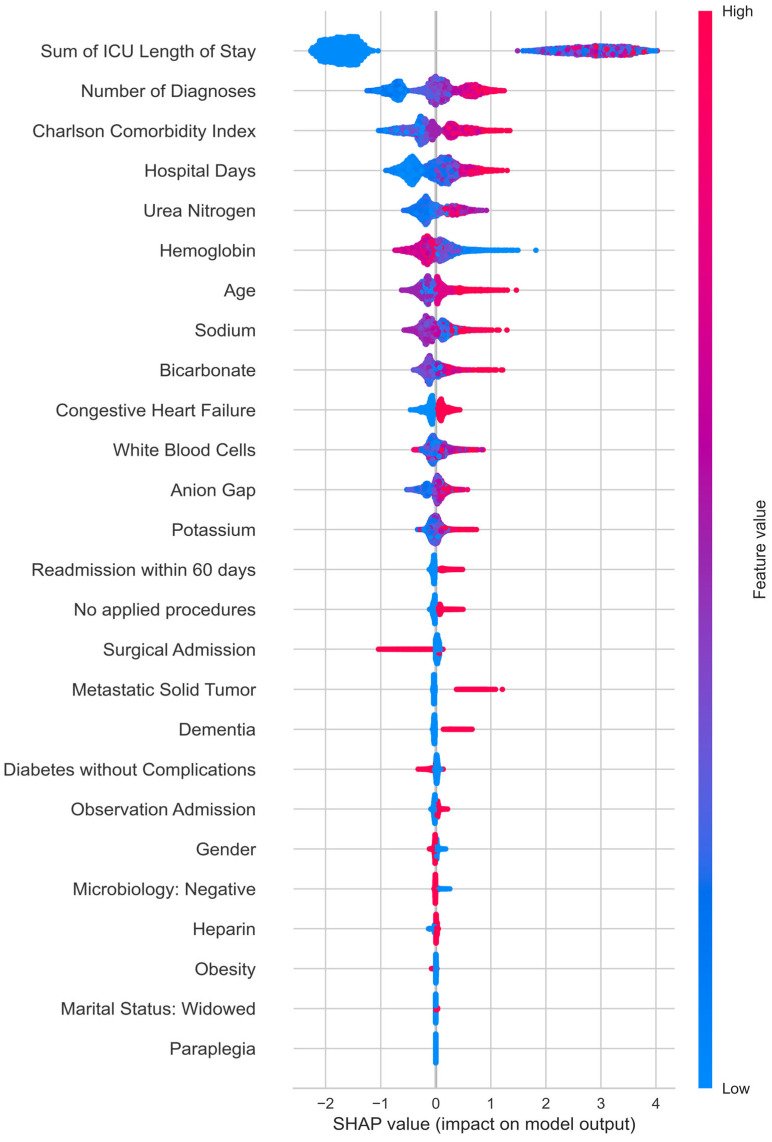
Beeswarm Plot: This plot visualizes the distribution of SHAP values across all features in the dataset, highlighting the impact of each feature on the model’s output.

**Figure 3 jcm-14-03697-f003:**
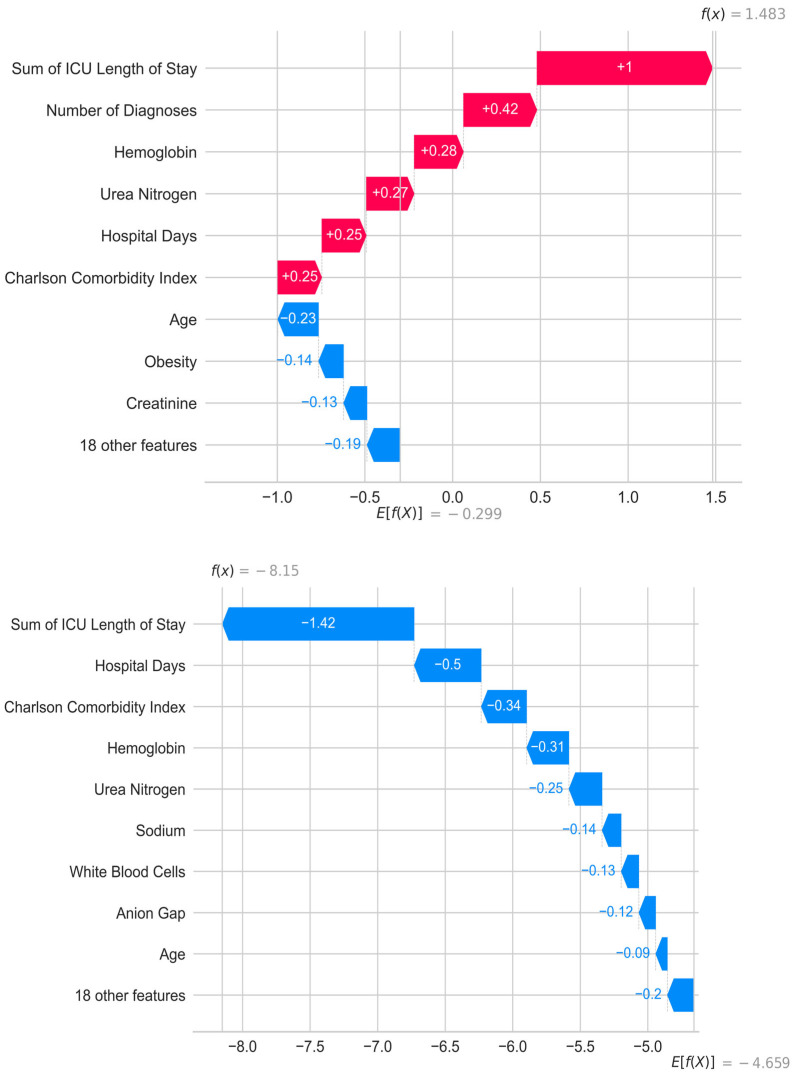
Waterfall Plot: This plot provides a detailed breakdown of SHAP values for two individual patients with different prediction outcomes. The first patient was classified by the model as high risk for mortality, while the second patient was predicted to be at low risk.

**Table 1 jcm-14-03697-t001:** Statistical analysis of 27 features used for building model.

Feature	Survivors (n = 6086)	Non-Survivors (n = 345)	*p*-Value (Two-Sided)
Age [years], median (Q1,Q3)	76 (68,84)	81 (73,88)	0.00000
Gender, male, n (%)	3416 (56.13%)	167 (48.41%)	0.00589
Congestive Heart Failure, n (%)	2768 (45.48%)	243 (70.43%)	0.00000
Dementia, n (%)	398 (6.54%)	49 (14.20%)	0.00000
Diabetes without Complications, n (%)	1313 (21.57%)	77 (22.32%)	0.79510
Paraplegia, n (%)	316 (5.19%)	54 (15.65%)	0.00000
Metastatic Solid Tumor, n (%)	192 (3.15%)	39 (11.30%)	0.00000
Readmission within 60 days, n (%)	1249 (20.52%)	120 (34.78%)	0.00000
Marital Status: Widowed, n (%)	1051 (17.27%)	74 (21.45%)	0.05546
Observation Admission, n (%)	2702 (44.40%)	103 (29.86%)	0.00000
Surgical Admission, n (%)	418 (6.87%)	5 (1.45%)	0.00012
Microbiology: Negative, n (%)	5113 (84.01%)	191 (55.36%)	0.00000
No applied procedures, n (%)	2197 (36.10%)	53 (15.36%)	0.00000
Heparin, n (%)	3984 (65.46%)	313 (90.72%)	0.00000
Obesity, n (%)	831 (13.65%)	37 (10.72%)	0.14204
Anion Gap [mEq/L], mean ± SD	15.89 ± 3.55	18.33 ± 4.27	0.00000
Bicarbonate [mEq/L], mean ± SD	27.79 ± 4.04	30.58 ± 5.30	0.00000
Creatinine [mg/dL], mean ± SD	1.43 ± 1.28	2.05 ± 1.87	0.00000
Hemoglobin [g/dL], mean ± SD	12.45 ± 2.07	11.6 ± 2.08	0.00000
Potassium [mEq/L], mean ± SD	4.69 ± 0.71	5.26 ± 0.98	0.00000
Sodium [mEq/L], mean ± SD	141.88 ± 4.01	144.94 ± 5.67	0.00000
Urea Nitrogen [mg/dL], mean ± SD	30.76 ± 21.11	48.97 ± 27.20	0.00000
White Blood Cells [10^9^/μL], mean ± SD	12.3 ± 9.55	17.35 ± 10.19	0.00000
Sum of ICU Length of Stay [days], median (Q1,Q3)	0 (0, 1.45)	3.19 (1.68, 7.73)	0.00000
Number of Diagnoses, median (Q1,Q3)	17 (12,22)	27 (21,33)	0.00000
Charlson Comorbidity Index, median (Q1,Q3)	5 (4,7)	8 (6,10)	0.00000
Hospital Days [days], median (Q1,Q3)	5 (2,9)	13 (8,23)	0.00000

**Table 2 jcm-14-03697-t002:** Performance metrics on training dataset. Results of cross-validation for five different machine learning algorithms.

Algorithm	AUC (95%CI)	Accuracy (95%CI)	Sensitivity (95%CI)	Specificity (95%CI)	MCC (95%CI)
LightGBM	0.873 (0.865–0.881)	0.862 (0.856–0.867)	0.886 (0.870–0.901)	0.860 (0.854–0.866)	0.439 (0.427–0.450)
XGBoost	0.871 (0.863–0.879)	0.846 (0.840–0.853)	0.899 (0.882–0.915)	0.843 (0.836–0.850)	0.422 (0.411–0.433)
RF	0.862 (0.854–0.870)	0.853 (0.846–0.859)	0.873 (0.855–0.890)	0.851 (0.845–0.858)	0.419 (0.409–0.429)
AdaBoost	0.844 (0.839–0.849)	0.764 (0.745–0.783)	0.934 (0.908–0.961)	0.754 (0.733–0.775)	0.348 (0.340–0.357)
LR	0.795 (0.785–0.805)	0.770 (0.765–0.776)	0.822 (0.801–0.842)	0.767 (0.762–0.773)	0.301 (0.291–0.312)

**Table 3 jcm-14-03697-t003:** Performance metrics on test dataset. Predictions made by LightGBM and XGBoost.

Algorithm	AUC (95%CI)	Accuracy (95%CI)	Sensitivity (95%CI)	Specificity (95%CI)	MCC (95%CI)
LightGBM	0.886 (0.849–0.919)	0.862 (0.842–0.880)	0.913 (0.842–0.973)	0.859 (0.839–0.878)	0.450 (0.388–0.507)
XGBoost	0.873 (0.834–0.908)	0.849 (0.828–0.868)	0.899 (0.823–0.967)	0.846 (0.825–0.866)	0.425 (0.362–0.483)

## Data Availability

Data is available on reasonable request.

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
