# Peer review of "Predicting Mortality in Atrial Fibrillation Patients Treated with Direct Oral Anticoagulants: A Machine Learning Study Based on the MIMIC-IV Database"

_jcm, 2025, doi:10.3390/jcm14113697_

Round 1
Reviewer 1 Report
Comments and Suggestions for Authors
To strengthen the clinical applicability of the SHAP analysis, the manuscript would benefit from the inclusion of case-based examples. For instance, presenting two hypothetical patients—one flagged by the model as high-risk and the other as low-risk—could illustrate how individual features influence predictions. Walking through each patient’s profile using SHAP values would not only bring the interpretability framework to life but also help clinicians understand how the model reasons in a real-world context.
The handling of missing data also warrants closer attention. The authors mention that features with more than 10% missing values were excluded and that records with any remaining missing data were dropped. While this approach simplifies data preprocessing, it carries the risk of introducing selection bias by excluding potentially relevant patient subgroups. A brief sensitivity analysis or discussion of possible imputation methods (such as mean substitution, k-nearest neighbors, or multiple imputation) would help to contextualize this limitation and enhance methodological transparency. Although the SHAP visualizations are well executed, the manuscript lacks deeper clinical interpretation of key predictors. For example, while ICU length of stay and heparin administration emerge as important features, it remains unclear how these should be understood in clinical terms. Are these variables merely markers of illness severity, or do they reflect specific treatment pathways? Clarifying these relationships would aid in translating the model’s output into actionable insights for clinicians. In addition, the manuscript would benefit from a clearer description of data preprocessing procedures. Beyond binary encoding of categorical variables, it remains uncertain whether numeric features such as laboratory values were normalized or scaled before being input into the models. Since standardization can affect model performance, particularly in algorithms like logistic regression, this should be addressed explicitly in the methods section. To enrich their discussion, authors should report real world data regarding the safety of DOACs such as Rivaroxaban (doi: 10.23736/S2724-5683.24.06546-3) Finally, while the authors acknowledge the limited predictive value of existing clinical scoring systems such as HATCH and CHA₂DS₂-VASc, their argument would be more compelling if supported by quantitative comparisons, even if drawn from published literature. Including metrics like the ROC-AUC or calibration data from these traditional tools—alongside the model's performance—would help frame the added value of the machine learning approach in more concrete terms. This study tackles a clinically important and timely issue—predicting short-term mortality in patients with atrial fibrillation (AF) who are treated with direct oral anticoagulants (DOACs). By incorporating a range of machine learning algorithms, including LightGBM, XGBoost, and Random Forest, the authors apply a contemporary, data-driven methodology to a field traditionally guided by standardized clinical risk scores. A particularly valuable aspect of their approach is the integration of SHAP (Shapley Additive Explanations), which enhances the interpretability of the model. This not only provides transparency in how predictions are made but also improves the potential for real-world clinical application, such as guiding bedside decision-making. The use of the MIMIC-IV database is another clear strength. This dataset is rigorously curated, openly accessible, and offers a broad array of structured clinical variables, which enables robust and high-quality model development. In addition, the authors utilized LASSO (Least Absolute Shrinkage and Selection Operator) for feature selection—an effective technique to reduce dimensionality and prevent overfitting when working with large variable sets. This resulted in a concise and clinically meaningful feature set of 27 predictors. Moreover, the manuscript reflects a strong commitment to thorough model evaluation. The authors systematically compared five different machine learning models, reporting key performance metrics such as AUC, sensitivity, specificity, and Matthews Correlation Coefficient (MCC), all presented with 95% confidence intervals. The final model, LightGBM, showed excellent performance on the test dataset (AUC = 0.886, sensitivity = 0.913), indicating a high ability to distinguish between survivors and non-survivors. Given the clinical importance of identifying at-risk patients, the decision to emphasize sensitivity in the model’s optimization is both appropriate and justified.Author Response
- To strengthen the clinical applicability of the SHAP analysis, the manuscript would benefit from the inclusion of case-based examples. For instance, presenting two hypothetical patients—one flagged by the model as high-risk and the other as low-risk—could illustrate how individual features influence predictions. Walking through each patient’s profile using SHAP values would not only bring the interpretability framework to life but also help clinicians understand how the model reasons in a real-world context.
Figure 3 demonstrates SHAP values for individual patients.
- The handling of missing data also warrants closer attention. The authors mention that features with more than 10% missing values were excluded and that records with any remaining missing data were dropped. While this approach simplifies data preprocessing, it carries the risk of introducing selection bias by excluding potentially relevant patient subgroups. A brief sensitivity analysis or discussion of possible imputation methods (such as mean substitution, k-nearest neighbors, or multiple imputation) would help to contextualize this limitation and enhance methodological transparency.
Missing data are a common challenge in clinical datasets and can impact the validity and generalizability of predictive models. Several imputation strategies are available, including simple approaches such as mean or median substitution, as well as more sophisticated methods like k-nearest neighbors (KNN) imputation and multiple imputation by chained equations (MICE). Simple imputation methods are computationally efficient but may underestimate variability and distort associations between variables. KNN imputation leverages the similarity between observations, potentially preserving local data structure, but can be sensitive to outliers and the choice of k. MICE, an advanced iterative technique, models each variable with missing values as a function of other variables, generating multiple plausible datasets and thus accounting for imputation uncertainty. In this study, we explored multiple imputation using MICE in addition to listwise deletion. However, the application of MICE did not result in a meaningful improvement in model performance or alter the main findings. Given the lack of demonstrable benefit from imputation, we opted for listwise deletion to maximize methodological transparency and interpretability while minimizing the risk of introducing imputation-related bias. This approach is further supported by the absence of evidence for systematic differences between complete and incomplete cases.
- Although the SHAP visualizations are well executed, the manuscript lacks deeper clinical interpretation of key predictors. For example, while ICU length of stay and heparin administration emerge as important features, it remains unclear how these should be understood in clinical terms. Are these variables merely markers of illness severity, or do they reflect specific treatment pathways? Clarifying these relationships would aid in translating the model’s output into actionable insights for clinicians.
The selected features serve as descriptors of the patient's clinical status and should not be interpreted as independent markers equivalent to those obtained from clinical studies. Artificial intelligence models should be utilized by specialists who are capable of interpreting the presented results and using them as a supportive tool for assessing the clinical condition of a patient, merely highlighting important factors that may increase or decrease the risk of mortality in a given individual. Model indications, such as the identification of heparin use as significant, should not be regarded as recommendations for treatment modification. Instead, such findings should be interpreted as reflecting the clinical status of the patient, which may require specific therapeutic interventions based on current medical knowledge and established clinical guidelines.
- In addition, the manuscript would benefit from a clearer description of data preprocessing procedures. Beyond binary encoding of categorical variables, it remains uncertain whether numeric features such as laboratory values were normalized or scaled before being input into the models. Since standardization can affect model performance, particularly in algorithms like logistic regression, this should be addressed explicitly in the methods section.
All continuous variables were standardized prior to the application of Logistic Regression (LR) and the Least Absolute Shrinkage and Selection Operator (LASSO) algorithm.
- To enrich their discussion, authors should report real world data regarding the safety of DOACs such as Rivaroxaban (doi: 10.23736/S2724-5683.24.06546-3)
The use of DOACs in the treatment of patients with AF is not only associated with significant therapeutic benefits but has also been shown to be safe in numerous studies, particularly with respect to the risk of major bleeding and intracranial hemorrhage. For instance, the XANTUS study—a prospective, real-world registry including over 6,700 patients with AF treated with rivaroxaban—reported a major bleeding rate of 2.1 per 100 patient-years and confirmed a low incidence of fatal bleeding events. Similarly, recent data from an Italian multicenter registry demonstrated that DOACs, including rivaroxaban, were safe and effective in routine clinical practice, with no major bleeding complications observed during a 12-month follow-up period. These findings are further corroborated by large-scale analyses from Danish cohorts, which have shown that DOACs are associated with a lower risk of death and major bleeding compared to warfarin, supporting their favorable safety profile in real-world settings.
- Finally, while the authors acknowledge the limited predictive value of existing clinical scoring systems such as HATCH and CHA₂DS₂-VASc, their argument would be more compelling if supported by quantitative comparisons, even if drawn from published literature. Including metrics like the ROC-AUC or calibration data from these traditional tools—alongside the model's performance—would help frame the added value of the machine learning approach in more concrete terms.
Compared to the CHA₂DS₂-VASc score (mean 3.01, SD 1.318), which achieved an AUC of 0.639 (95% CI: 0.611–0.665) for mortality prediction, the results obtained by the presented models were substantially superior.
This study tackles a clinically important and timely issue—predicting short-term mortality in patients with atrial fibrillation (AF) who are treated with direct oral anticoagulants (DOACs). By incorporating a range of machine learning algorithms, including LightGBM, XGBoost, and Random Forest, the authors apply a contemporary, data-driven methodology to a field traditionally guided by standardized clinical risk scores. A particularly valuable aspect of their approach is the integration of SHAP (Shapley Additive Explanations), which enhances the interpretability of the model. This not only provides transparency in how predictions are made but also improves the potential for real-world clinical application, such as guiding bedside decision-making. The use of the MIMIC-IV database is another clear strength. This dataset is rigorously curated, openly accessible, and offers a broad array of structured clinical variables, which enables robust and high-quality model development. In addition, the authors utilized LASSO (Least Absolute Shrinkage and Selection Operator) for feature selection—an effective technique to reduce dimensionality and prevent overfitting when working with large variable sets. This resulted in a concise and clinically meaningful feature set of 27 predictors. Moreover, the manuscript reflects a strong commitment to thorough model evaluation. The authors systematically compared five different machine learning models, reporting key performance metrics such as AUC, sensitivity, specificity, and Matthews Correlation Coefficient (MCC), all presented with 95% confidence intervals. The final model, LightGBM, showed excellent performance on the test dataset (AUC = 0.886, sensitivity = 0.913), indicating a high ability to distinguish between survivors and non-survivors. Given the clinical importance of identifying at-risk patients, the decision to emphasize sensitivity in the model’s optimization is both appropriate and justified.
Reviewer 2 Report
Comments and Suggestions for Authors
The study contributes to the advancement of precision medicine by offering a high-accuracy tool for predicting mortality. However, further validation on diverse cohorts and integration into clinical workflows are required for practical implementation. To enhance the article:
-
Specify exclusion criteria (e.g., in-hospital mortality reasons) in the Methods section.
-
Clearly state the limitations of the study.
-
Clarify whether continuous variables were scaled before applying LASSO, as this could impact feature selection.
-
Statistically compare sensitivity and specificity across models.
-
Include traditional risk scores (e.g., CHA₂DS₂-VASc, HATCH) as baseline models and compare their AUC/Accuracy with LightGBM.
-
Perform temporal validation by splitting data by year to assess model robustness against data drift.
-
Discuss ethical implications of false-positive/false-negative predictions (e.g., overtreatment risks) in the Discussion section.
Author Response
- Specify exclusion criteria (e.g., in-hospital mortality reasons) in the Methods section.
- The exclusion criteria included cases with missing data and patients who experienced in-hospital death. “Patients who died during their hospital stay were excluded from the cohort.” & “Data cleaning was conducted by removing all variables with more than 10% missing values.” & “Patient records containing missing values were excluded from the dataset”
- Clearly state the limitations of the study.
- The model is not limited to predicting death from a specific cause due to the limited scope of the data. An additional limitation is the availability of data from only one fa-cility. Following the example of the MIMIC database, it would be beneficial for other large institutions to develop similar datasets. This would reduce data bias and improve the testing of developed models. Another important limitation is the use of a simplified preprocessing approach for handling missing data by removing incomplete records. While this reduces complexity and maintains a streamlined workflow, it introduces the risk of bias. Exploring the impact of various data imputation techniques would be an interesting direction for future research. Preliminary attempts to apply the Multiple Imputation by Chained Equations (MICE) method did not yield a significant improve-ment in results. Additionally, the study does not employ more complex algorithms or ensemble methods, which could potentially improve performance but often at the ex-pense of model interpretability.
- Clarify whether continuous variables were scaled before applying LASSO, as this could impact feature selection.
- All continuous variables were standardized prior to the application of Logistic Regression (LR) and the Least Absolute Shrinkage and Selection Operator (LASSO) algorithm
- Statistically compare sensitivity and specificity across models.
- The Z-test performed on the results obtained by the LightGBM and XGBoost models on the test data did not reveal statistically significant differences in sensitivity or specificity (p >> 0.05).
- Include traditional risk scores (e.g., CHA₂DS₂-VASc, HATCH) as baseline models and compare their AUC/Accuracy with LightGBM.
- Compared to the CHA₂DS₂-VASc score (mean 3.01, SD 1.318), which achieved an AUC of 0.639 (95% CI: 0.611–0.665) for mortality prediction, the results obtained by the presented models were substantially superior. In light of the significant class imbalance, comparing accuracy is less relevant.
- Perform temporal validation by splitting data by year to assess model robustness against data drift.
- “All dates in MIMIC-IV have been deidentified by shifting the dates into a future time period between 2100 - 2200. This shift is done independently for each patient, and as a result two patients admitted in the deidentified year 2120 cannot be assumed to be admitted in the same year.” Splitting by deidentified year could lead to biased results because the grouping is arbitrary and unrelated to the actual timeline of patient admissions. This could distort the evaluation of model performance and fail to provide meaningful insights into its robustness against real-world data drift.
- Discuss ethical implications of false-positive/false-negative predictions (e.g., overtreatment risks) in the Discussion section.
False-positive predictions may lead to unnecessary interventions, such as increased monitoring, additional medications, or invasive procedures. These actions could expose patients to avoidable side effects, increased healthcare costs, and psychological distress due to being labeled as high-risk. Overestimating mortality risk could divert limited healthcare resources (e.g., ICU beds or specialist attention) away from patients who genuinely need them, potentially compromising care for others. On the other hand, false-negative predictions may result in high-risk patients being overlooked for neces-sary interventions, leading to preventable adverse outcomes, including death. This undermines the primary goal of the predictive model. If patients or clinicians perceive the model as unreliable, it could erode trust in AI-based tools, hindering their adoption in clinical practice. Ethical considerations demand a careful balance between sensitivity (minimizing false negatives) and specificity (minimizing false positives). For mortality prediction, prioritizing sensitivity may be more ethical to ensure high-risk patients are not missed, but this must be weighed against the risks of overtreatment. Predictions should be used to support, not replace, clinical judgment. Patients must be informed about the limitations of the model, including the possibility of false predictions, to en-sure they can make autonomous decisions about their care. False predictions may dis-proportionately affect certain patient groups if the model is biased due to imbalanced training data. This raises ethical concerns about fairness and equity in healthcare de-livery.
Round 2
Reviewer 1 Report
Comments and Suggestions for Authors
Congratulatuons to the authors for having modified their manuscript according to my comments.
Reviewer 2 Report
Comments and Suggestions for Authors
Accept in present form